# Effectiveness of the BNT162b2 mRNA Vaccine Compared with Hybrid Immunity in Populations Prioritized and Non-Prioritized for COVID-19 Vaccination in 2021–2022: A Naturalistic Case-Control Study in Sweden

**DOI:** 10.3390/vaccines10081273

**Published:** 2022-08-07

**Authors:** Armin Spreco, Örjan Dahlström, Anna Jöud, Dennis Nordvall, Cecilia Fagerström, Eva Blomqvist, Fredrik Gustafsson, Jorma Hinkula, Thomas Schön, Toomas Timpka

**Affiliations:** 1Department of Health, Medicine, and Caring Sciences, Linköping University, 58183 Linköping, Sweden; 2Regional Executive Office, Region Östergötland, 58225 Linköping, Sweden; 3Department of Behavioral Sciences and Learning, Linköping University, 58183 Linköping, Sweden; 4Department of Laboratory Medicine, Lund University, 22100 Lund, Sweden; 5Department of Research and Education, Skåne University Hospital, 21421 Lund, Sweden; 6Qulturum Development Department, Region Jönköping County, 55592 Jönköping, Sweden; 7Department of Research, Region Kalmar County, 39185 Kalmar, Sweden; 8Department of Computer and Information Science, Linköping University, 58183 Linköping, Sweden; 9Department of Electrical Engineering, Linköping University, 58183 Linköping, Sweden; 10Department of Biomedical and Clinical Sciences, Linköping University, 58183 Linköping, Sweden; 11Department of Infectious Diseases, County of Östergötland and Kalmar, Linköping University, 58183 Linköping, Sweden

**Keywords:** COVID-19, vaccination program, hybrid immunity, case-control study design, effectiveness, epidemiology

## Abstract

The term hybrid immunity is used to denote the immunological status of vaccinated individuals with a history of natural infection. Reports of new SARS-CoV-2 variants of concern motivate continuous rethought and renewal of COVID-19 vaccination programs. We used a naturalistic case-control study design to compare the effectiveness of the BNT162b2 mRNA vaccine to hybrid immunity 180 days post-vaccination in prioritized and non-prioritized populations vaccinated before 31 July 2021 in three Swedish counties (total population 1,760,000). Subjects with a positive SARS-CoV-2 test recorded within 6 months before vaccination (*n* = 36,247; 6%) were matched to vaccinated-only controls. In the prioritized population exposed to the SARS-CoV-2 Alpha and Delta variants post-vaccination, the odds ratio (OR) for breakthrough infection was 2.2 (95% CI, 1.6–2.8; *p* < 0.001) in the vaccinated-only group compared with the hybrid immunity group, while in the later vaccinated non-prioritized population, the OR decreased from 4.3 (95% CI, 2.2–8.6; *p* < 0.001) during circulation of the Delta variant to 1.9 (95% CI, 1.7–2.1; *p* < 0.001) with the introduction of the Omicron variant (B.1.617.2). We conclude that hybrid immunity provides gains in protection, but that the benefits are smaller for risk groups and with circulation of the Omicron variant and its sublineages.

## 1. Introduction

Reports on waning immunity and the discovery of new SARS-CoV-2 variants of concern motivate continuous rethought and renewal of COVID-19 vaccination programs [1,2,3,4]. The term hybrid immunity is used to denote the immunological status of vaccinated individuals with a history of natural infection [5]. At the present stage of the COVID-19 pandemic, when large factions of the global population have been infected at least once [6,7,8], detailed knowledge on hybrid immunity taking into account the different SARS-CoV-2 variants is needed for planning of effective vaccination programs.

The BNT162b2 vaccine, developed using an mRNA-based platform, is the major vaccine used so far in the Swedish COVID-19 vaccination program [9,10]. At the time of rollout in early 2021, the vaccine had been shown to produce a protective B cell response, and to some extent, T cell responses [11,12] and prevent hospitalization and death among those with the ancestral SARS-CoV-2 strain and the Alpha (B.1.1.7) variant circulating at that time [13]. The effectiveness of the vaccine in the Swedish program was contested when the Delta variant of concern (B.1.617.2) replaced the Alpha variant as the dominant SARS-CoV-2 variant in August 2021 [14], causing severe disease and devastating pressure on intensive care units. Laboratory studies had indicated reduced sensitivity of this variant to antibody neutralization [15], but at that time, only minor differences in vaccine effectiveness had been reported for any of the initial variants of concern [16]. The Omicron variant (B.1.1.529) was first detected in Sweden in December 2021. This variant was predicted to be more transmissible [17] and infectious [18] than the initial variants, but with a less severe clinical course. These assumptions were confirmed when the number of Omicron cases increased rapidly to dominate the dissemination of SARS-CoV-2 in Sweden from mid-December 2021 [14].

In Sweden (population 10.4 million), 2.6 million cases of SARS-CoV-2 infection had been registered by May 2022 [14]. We aimed to compare the effectiveness of the BNT162b2 mRNA vaccine with hybrid immunity 180 days post-vaccination in population groups prioritized and not prioritized for vaccination. The study was performed in a naturalistic setting in three Swedish counties (total population 2.2 million) taking circulating SARS-CoV-2 variants and vaccination schedules into account, including administration of booster doses.

## 2. Materials and Methods

A case-control design was used for the study, which was performed while the mass SARS-CoV-2 testing policy was in effect in Sweden during 2021–2022. Hybrid immunity cases were defined as subjects vaccinated twice with the BNT162b2 vaccine, and who had a SARS-CoV-2 infection recorded within 6 months before vaccination. Subjects with hybrid immunity were matched with regards to age, sex, and week of vaccination to vaccinated controls who had not been infected within 6 months before vaccination. SARS-CoV-2 infection was defined as a positive reverse-transcription (RT) polymerase chain reaction (PCR) test. The primary outcome measure was a SARS-CoV-2 infection within 180 days post-vaccination (breakthrough infection) counted from 14 days after the second dose of vaccine. Secondary outcome measures were breakthrough infections within 60, 90, 120, and 150 days post-vaccination.

### 2.1. Study Population

The primary population consisted of adults aged 18 years or older in Östergötland, Jönköping, and Skåne counties, Sweden (*n* = 1,760,000) who had received the standard two-dose vaccination with the BNT162b2 vaccine by 31 July 2021 (*n* = 576,526; 32.8%). We divided this population by vaccination phase into prioritized and non-prioritized populations. The prioritized population, vaccinated in a first phase, included the elderly, individuals in medical risk groups, and health care workers; the non-prioritized population, vaccinated in a second phase, mainly included middle-aged individuals without medical risks. The prioritized population received the second dose between weeks 3 and 22, 2021, and the non-prioritized population received the second dose of vaccine between weeks 23 and 31, 2021.

### 2.2. Data Collection

Individual-level vaccination and PCR test data were retrieved from the county-wide health information systems [19]. Aggregated data on circulating SARS-CoV-2 variants were obtained from the Swedish Public Health Agency (FoHM) [14] and verified against data from the local laboratories. In the primary population, 36,247 individuals (6.6%) had acquired hybrid immunity through vaccination by 31 July 2021. We matched these cases of hybrid immunity to vaccinated controls 1:1 with respect to the same sex, age, and week for the second dose of vaccine to form the case-control population.

### 2.3. Data Analysis

Descriptive presentations were prepared separately for the primary and case-control populations. These populations were further divided into subpopulations by vaccination phase. The incidence of positive PCR tests for SARS-CoV-2 from 1 September 2020 to 10 February 2022 (corresponding to 6 months before receiving the first dose of vaccine to 14 days after the final second dose, including 180 days of follow-up from 31 July 2021) is displayed using descriptive statistics, highlighting the dominant variant of SARS-Cov-2. 

Multiple binary logistic regression models with breakthrough infection (no = 0, yes = 1) within 60, 90, 120, 150, and 180 days post-vaccination as outcomes were applied on the case-control population with hybrid immunity (yes = 0, no = 1) as the explanatory variable and booster dose received during post-vaccination follow-up (yes = 0, no = 1) as the adjustment variable. The variables used to match the cases with the controls (sex, age, and week of vaccination) were also included in the models. Individuals who died before breakthrough infection during the follow-up period were censored. The analyses were performed using the Statistical Package for the Social Sciences (SPSS) for Windows (version 28.0).

## 3. Results

### 3.1. Vaccination Patterns, Exposures, and Incidences

In the total population vaccinated by 31 July 2021 (*n* = 576,526), 66% received their second dose of vaccine in the prioritized phase and 34% in the non-prioritized phase (Appendix A). The Alpha variant of SARS-CoV-2 dominated dissemination in the study counties from January to July 2021 (Figure 1). The Delta variant gained dominance in August 2021, but was replaced by Omicron (sublineages BA.1 and BA.2) as the dominating variant from late December 2021 to the end of the study period in February 2022. The prioritized population was only exposed to the Alpha and Delta variants within 6 months post-vaccination, whereas the non-prioritized population was first exposed to the Delta variant and then the Omicron variant in the final phase of the 6-month follow-up. In the total vaccinated population, 2.0% of individuals had a breakthrough infection recorded in the 180-day period post-vaccination. In the prioritized population, 0.6% of individuals had a breakthrough infection; 0.4% of those with hybrid immunity and 0.6% with vaccine alone. In the non-prioritized population, 4.7% of individuals had a breakthrough infection; 2.8% of those with hybrid immunity and 5.0% with vaccination alone.

### 3.2. Case-Control Population

Almost all cases were matched exactly to controls (i.e., the same gender, age, and week of vaccination); 0.1% (*n* = 29) had no exact match. The cases with no match and their controls were excluded from the analyses (*n* = 58). After excluding individuals who died before breakthrough infection during the follow-up period (*n* = 394), the final case-control study population consisted of 76,042 individuals. In this population, 47% received their vaccine in the prioritized population and 53% in the non-prioritized population (Table 1). During the 180-day follow-up period, 66% of the prioritized population and 53% of the non-prioritized population subpopulation received a booster dose of vaccine. In total, 2.4% of the case-control population had a breakthrough infection recorded in the post-vaccination period.

### 3.3. Breakthrough Infections in the Case-Control Population

In the prioritized population exposed to the initial SARS-CoV-2 variants, 0.7% had a breakthrough infection; 0.4% of individuals with hybrid immunity and 0.9% with vaccination alone. The adjusted odds ratio for breakthrough infection adjusted for booster dose, age, sex, and week of vaccination among the controls compared with cases with hybrid immunity increased from 1.0 (95% confidence interval [CI], 0.6–1.5; *p* = 0.822) 60 days after vaccination to 2.2 (95% CI, 1.6–2.8; *p* < 0.001) 180 days after vaccination (Figure 2). In the non-prioritized population vaccinated in the second phase who were also exposed to the Omicron variant, 3.9% had a breakthrough infection; 2.8% of individuals with hybrid immunity and 5.1% with vaccination alone. The adjusted odds ratio for breakthrough infection among the controls was 4.3 (95% CI, 2.2–8.6; *p* < 0.001) at 60 days after vaccination when the Delta variant dominated, but it decreased from day 120, when the Omicron variant was introduced, to 1.9 (95% CI, 1.7–2.1; *p* < 0.001) 180 days after vaccination (Figure 3).

### 3.4. Booster Doses of Vaccine

In the prioritized population, receiving a booster dose of vaccine during the follow-up period was highly protective against breakthrough infection. The odds ratio for breakthrough infection adjusted for hybrid immunity, age, sex, and vaccination week for individuals who had not received a booster dose was 27 (95% CI, 11–66; *p* < 0.001) compared with those who had received a booster dose. Individuals with hybrid immunity who had not received a booster dose had an unadjusted odds ratio of 17 (simplified analysis due to few events) for breakthrough infection, compared with those who had received a booster dose. Those with vaccination alone who had not received a booster dose had an unadjusted odds ratio of 38 (simplified analysis due to few events) for breakthrough infection. In addition, in the non-prioritized population, receiving a booster dose of vaccine was inversely associated with breakthrough infection. Individuals who had not received a booster dose had an adjusted odds ratio of 8.4 (95% CI, 7.3–9.8; *p* < 0.001) for breakthrough infection. The individuals with hybrid immunity who had not received a booster dose had an adjusted odds ratio of 9.4 (95% CI, 7.7–12; *p* < 0.001) for breakthrough infection, compared with those who had received a booster dose; the corresponding odds ratio for individuals with vaccination alone was 7.1 (95% CI, 5.7–9.0; *p* < 0.001).

## 4. Discussion

This study set out to compare the effectiveness of the BNT162b2 mRNA vaccine to hybrid immunity in a naturalistic setting in Sweden between January 2021 and February 2022. In a first vaccinated prioritized population exposed to the initial SARS-CoV-2 variants post-vaccination, we found 2 months after vaccination that the vaccinated-only controls did not show an increased likelihood for breakthrough infection compared with cases of hybrid immunity but that, at the end of the 6-month follow-up period, the vaccinated-only controls had a two times higher odds ratio. In the later vaccinated non-prioritized population, the vaccinated-only controls had a four times higher odds ratio for breakthrough infection, compared with the prioritized population, while exposed to the Delta variant, but this difference decreased with the introduction of the Omicron variant (sublineages BA.1 and BA.2). Receiving a booster dose was associated with a notable protective effect in the prioritized population; booster doses in the non-prioritized population exposed to the Omicron variant had one third of that impact.

In the first vaccinated prioritized population exposed to the initial SARS-CoV-2 variants, few breakthrough infections (0.7%) were recorded during the 180-day follow-up period. Nonetheless, the odds ratio for breakthrough infection in the vaccinated-only controls was two times that for the hybrid immunity cases. These results are in agreement with previous reports, e.g., a study from Qatar that reported a lower risk for breakthrough infections in the hybrid immunity group (hazard ratio, 0.2; 95% CI, 0.2–0.2) at 120 days of follow-up [20]. Our data also indicated earlier waning of protection from the mRNA vaccine alone compared with hybrid immunity. Previous studies have suggested that immunity from SARS-CoV-2 infection lasts from 5 to 12 months, whereas protection from mRNA vaccines begins to wane from 3 months [21,22,23,24,25]. In patients with COVID-19, virus-specific CD4^+^ and CD8^+^ T cell response (with SARS-CoV-2-conserved T cell epitopes detected in the S1-spike/RBD proteins) has been seen in patients with less severe illness, possibly reflecting cellular immunity suppressing SARS-CoV-2-associated disease [26,27,28]. With regard to the immunological mechanisms that could explain our observations regarding waning of protection against the initial variants, a recent investigation [29] comparing protection from mRNA vaccine and hybrid immunity reported higher numbers of RBD-specific memory B cells and SARS-CoV-2 variant-neutralizing antibodies in individuals with hybrid immunity, but also reported that this difference was reduced by booster doses of vaccine. Moreover, memory CD4^+^ T cells in individuals with hybrid immunity exhibited a cytokine-producing profile that could not be achieved by the booster doses. The longer duration of protection for those with hybrid immunity observed in our study may thus be explained hypothetically by induction of SARS-CoV-2-specific memory lymphocytes. According to current immunological evidence, the reduced risk of breakthrough infection after booster doses of vaccine can be explained by improved humoral immunity. These observations are in concordance with the fact that the population-level protective effect from mRNA vaccines is mainly mediated through short-term humoral immunity.

In the later vaccinated non-prioritized population exposed also to the Omicron variant, an almost six times larger proportion (3.9%) of breakthrough infections was recorded than in the prioritized population; in 2.8% of individuals with hybrid immunity and in 5.1% of those with immunity from vaccination alone. While exposed to the Delta variant post-vaccination, the odds ratio for breakthrough infection for vaccinated-only controls in the non-prioritized population was up to four times higher compared with the controls in the prioritized population. The reason for the higher odds ratio may be stronger protection from natural immunity in younger individuals; a study of Danish population data from the first two pandemic waves in 2020 estimated that such protection was substantial (80–83%) in people younger than 65 years, while it was more moderate (47%) among those aged 65 years and older [24]. Regarding the reduction in the odds ratio observed with the introduction of the Omicron variant, the general protection from mRNA vaccines against this variant has been found to be weaker than against the initial variants [30,31], while a French population-based study [32] reported that hybrid immunity against Omicron acquired through infection with one of the initial variants reached 67% compared with 43% from vaccination alone. Corresponding hybrid immunity against the Delta variant was reported to reach above 90%. Moreover, for the Omicron variant, booster doses of mRNA vaccine have been reported to enhance levels of neutralizing antibodies, but the responses remain four to six times lower than to the vaccine strain spike protein [33,34,35,36]. In our study, we found that the protection from booster doses was lower against the Omicron variant than against the initial variants and that hybrid immunity status did not enhance the protection from booster doses against the Omicron variant. Thus, our epidemiological findings confirm previous observations regarding the distinctive nature of the Omicron variant. The variant (B.1.1.529) includes five sublineages, BA.1, BA.2, BA.3, BA.4., and BA.5 [37]. The lineages circulating in Sweden during our study were BA.1 and BA.2, while BA.4, BA.5, and sublineage BA.2.12.1 became dominant in Europe and the USA later in 2022 [38]. These observations add to concerns that immunity acquired through infection with SARS-CoV-2 or booster doses of vaccine based on the ancestral variant or combinations involving Omicron BA.1 components may become increasingly ineffective [39].

This study has strengths and limitations that should be taken into consideration when interpreting the results. It is one of the first studies that specifically addresses the importance of hybrid immunity in large-scale COVID-19 vaccination programs. However, investigations of hybrid immunity in vaccination programs are restricted to observational designs with inherent limitations. Our results must therefore be interpreted considering potential bias, e.g., different testing frequencies among cases and controls, misclassifications due to unrecognized infections, and imperfect sensitivity and specificity of PCR tests [5]. We did not compare the non-vaccinated immunity status with vaccination alone, or hybrid immunity, due to the risk of confounding by factors that differ between the non-vaccinated and vaccinated groups, e.g., socioeconomic status and living in institutionalized settings. In particular, health consciousness and trust in authorities may differ between vaccinated and non-vaccinated groups and may also affect other preventive behaviors. Moreover, the incidence of recorded breakthrough infections varied during the post-vaccination follow-up periods. Large differences observed using relative measures may correspond to small differences in absolute numbers. Moreover, attention should be paid to the fact that we did not determine the specific SARS-CoV-2 variants at the individual level. However, the data from national and regional laboratories (not reported) on which our classification of SARS-CoV-2 variants was based indicate that the shifts between dominating variants occurred rapidly (in about 2 weeks). This suggests that the classification into time periods should be sufficiently accurate for the purpose of this study. Finally, we did not include hospitalizations and deaths in the study because the number of hybrid immunity cases with a recorded event during the 180-day follow-up period was considered too few to analyze.

## 5. Conclusions

We found that hybrid immunity provided gains in effectiveness, compared with BNT162b2 mRNA vaccine alone, in a naturalistic vaccination program setting during 2021–2022, but that the benefit was smaller for risk groups. Both hybrid immunity and the vaccine alone provided less effective protection against the Omicron variant (sublineages BA.1 and BA.2), leading to higher incidences of breakthrough infections. Responding to the increasing circulation of novel Omicron sublineages, the WHO introduced in June 2022 the Variant of Concern Lineages Under Monitoring (VOC-LUM) categorization [40]. In this rapidly changing SARS-CoV-2 immunity landscape, continuous population-level surveillance of infections and vaccination coverage is required to support effective vaccination program scheduling. 

## Figures and Tables

**Figure 1 vaccines-10-01273-f001:**
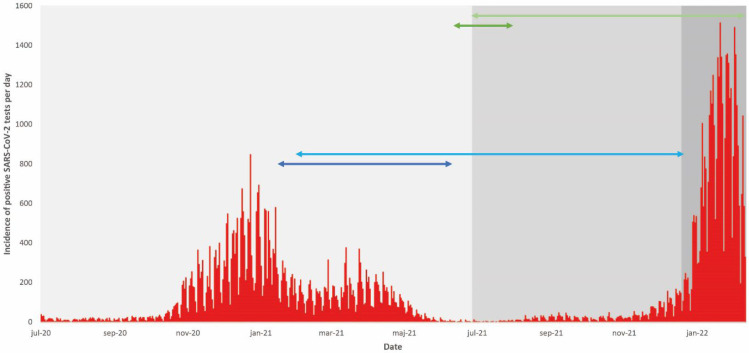
Incidence of positive SARS-CoV-2 tests (red stacks) in the total vaccinated population during the study period. The dominating SARS-CoV-2 variant of concern (light grey, grey, and dark grey areas representing the periods dominated by the Alpha, Delta and Omicron variants, respectively) is indicated, as well as the time periods for vaccination and follow-up for the prioritized risk population (dark blue line and light blue line, respectively) and the non-prioritized population (dark green line and light green line, respectively).

**Figure 2 vaccines-10-01273-f002:**
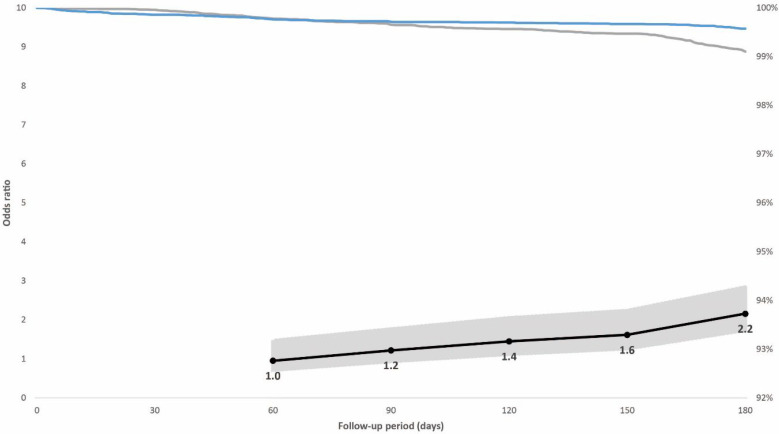
Individuals in the case-control population vaccinated in the prioritized first phase. Right vertical axis: distribution of breakthrough infections in the 180-day follow-up period displayed by immunity status at baseline (blue line: hybrid immunity, grey line: vaccination only). Left vertical axis: odds ratios (black line) including 95% confidence intervals (grey-shaded area) for breakthrough infections adjusted for booster dose administration among individuals with vaccination alone compared with individuals with hybrid immunity for post-vaccination days 60, 90, 120, 150, and 180. The initial variants of concern (Alpha and Delta) dominated SARS-CoV-2 circulation during the post-vaccination follow-up period.

**Figure 3 vaccines-10-01273-f003:**
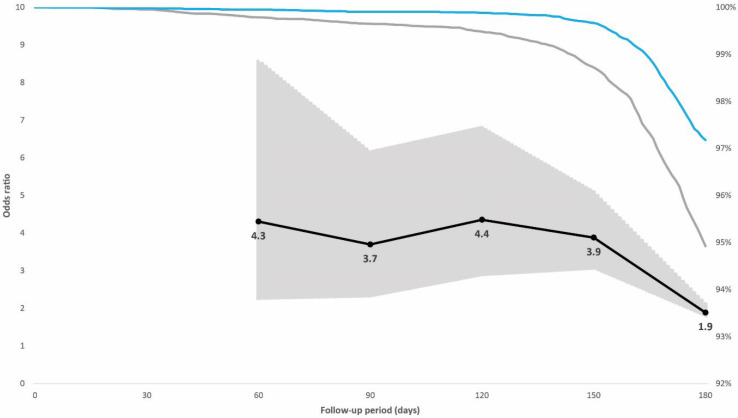
Individuals in the case-control population vaccinated in the non-prioritized second phase. Right vertical axis: distribution of breakthrough infections in the 180-day follow-up period displayed by immunity status at baseline (blue line hybrid immunity, grey line vaccination only). Left vertical axis: odds ratios (black line) including 95% confidence intervals (grey-shaded area) for breakthrough infections adjusted for booster dose administration among individuals with vaccination alone compared with individuals with hybrid immunity for post-vaccination days 60, 90, 120, 150, and 180. The Omicron variant replaced Delta as the dominant variant in the weeks before and after day 120 of the follow-up period.

**Table 1 vaccines-10-01273-t001:** Age and sex in the case-control population vaccinated with the BNT162b2 vaccine by 31 July 2021 in Östergötland, Jönköping, and Skåne counties, Sweden (*n* = 76,042) displayed by vaccination phase (prioritized first phase and non-prioritized second phase).

Age	Women		Men		Total	
	*n*	%	*n*	%	*n*	%
18–39 years	7422	10	3766	5	11,188	15
Prioritized	3716	5	1547	2	5263	7
Non-prioritized	3706	5	2219	3	5925	8
40–64 years	24,187	32	22,210	29	46,397	61
Prioritized	9564	13	5478	7	15,042	20
Non-prioritized	14,623	19	16,732	22	31,355	41
65–79 years	6472	8	7021	9	13,493	18
Prioritized	5129	7	5636	7	10,765	14
Non-prioritized	1343	2	1385	2	2728	4
80+ years	3149	4	1815	2	4964	6
Prioritized	3026	4	1740	2	4766	6
Non-prioritized	123	0	75	0	198	0
Total	41,230	54	34,812	46	76,042	100
Prioritized	21,435	28	14,401	19	35,836	47
Non-prioritized	19,795	26	20,437	27	40,206	53

## Data Availability

No data available.

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
