# Peer review of "Effectiveness of the BNT162b2 mRNA Vaccine Compared with Hybrid Immunity in Populations Prioritized and Non-Prioritized for COVID-19 Vaccination in 2021–2022: A Naturalistic Case-Control Study in Sweden"

_vaccines, 2022, doi:10.3390/vaccines10081273_

Round 1
Reviewer 1 Report
The case-control study proposed by Armin et.al addresses the effectiveness of the BNT162b2 mRNA vaccine and highlights the hybrid immunity factor. Although the manuscript has been written nicely, here are some of my concerns listed below:
1. Through no fault of researchers, the data will be limited when using patients from particular populations. Factors such as access to healthcare and what vaccines were available and when they were available can differ between populations. They may affect the results, meaning they mightn’t be applicable to all groups.
2. The fact that many of these investigations were conducted before omicron, which has a severely mutated spike protein, is a serious constraint. These modifications lessen its susceptibility to the antibodies produced by earlier infections or vaccinations (all of which are based on the original covid strain). In fact, research shows that compared to waves generated by previous variations, the reinfection rate during the omicron wave was substantially greater. Therefore, outcomes from the following research might be significantly different.
3. There are still many aspects of immunology that we are still working to completely understand. But because the hazards of infection greatly outweigh those of vaccination, we won't be looking for a "natural" infection to increase my protection against COVID.
4. The authors have mentioned that they have characterized the patient groups into prioritized and non-prioritized where elderly and health workers are considered as prioritized and middle-aged individuals are non-prioritized. But in table 1, the elderly population e.g. 80+ years also have non-prioritized individuals. Can you justify why you have taken the elderly population as non-prioritized?
5. Since there are many limitations reported in your study line 282-304, why the statement "we conclude" has been reported. Rather it should be like " The results strengthen the evidence-based rationale....."
6. There is a good case-control study in Sweden previously reported by https://pubmed.ncbi.nlm.nih.gov/35131043/. Please have a look and cite it.
Author Response
Reviewer 1
The case-control study proposed by Armin et.al addresses the effectiveness of the BNT162b2 mRNA vaccine and highlights the hybrid immunity factor. Although the manuscript has been written nicely, here are some of my concerns listed below:
1. Through no fault of researchers, the data will be limited when using patients from particular populations. Factors such as access to healthcare and what vaccines were available and when they were available can differ between populations. They may affect the results, meaning they mightn’t be applicable to all groups.
Authors’ response: Thank you! We agree. The study used a naturalistic design with the ambition to cover the practical aspects associated with provision of a COVID-19 vaccination program to a large population. The generalization of the results to other settings must be done with extreme caution, which also is pointed out in the limitations section,
- The fact that many of these investigations were conducted before omicron, which has a severely mutated spike protein, is a serious constraint. These modifications lessen its susceptibility to the antibodies produced by earlier infections or vaccinations (all of which are based on the original covid strain). In fact, research shows that compared to waves generated by previous variations, the reinfection rate during the omicron wave was substantially greater. Therefore, outcomes from the following research might be significantly different.
Authors’ response: Thank you! We again agree. The omicron VoC and its different sublineages indeed require separate and close attention. However, the present study was extended as long as the mass testing directive was operational in Sweden (until February 2022) and thus covered the first part of the omicron wave (with the BA.1 lineage circulating). This allowed us to calculate some initial comparisons with the previous VoCs. These analyses are reported in the paper. We also express the need for continued separate investigations of the omicron VoC and its sublineages in the future.
- There are still many aspects of immunology that we are still working to completely understand. But because the hazards of infection greatly outweigh those of vaccination, we won't be looking for a "natural" infection to increase my protection against COVID.
Authors’ response: Thank you! We again fully agree.
- The authors have mentioned that they have characterized the patient groups into prioritized and non-prioritized where elderly and health workers are considered as prioritized and middle-aged individuals are non-prioritized. But in table 1, the elderly population e.g. 80+ years also have non-prioritized individuals. Can you justify why you have taken the elderly population as non-prioritized?
Authors’ response: Thank you! This thus was a naturalistic study. Small minorities of the formally prioritized groups were for different reasons (other illness, travels, etc.) not vaccinated in the first phase. This was the reason for that a small number of elderly were vaccinated in the later phase.
- Since there are many limitations reported in your study line 282-304, why the statement "we conclude" has been reported. Rather it should be like " The results strengthen the evidence-based rationale.....”
Authors’ response: Thank you, a revision has been made according to your suggestion and the requests from the editor.
- There is a good case-control study in Sweden previously reported by https://pubmed.ncbi.nlm.nih.gov/35131043/. Please have a look and cite it.
Authors’ response: Thank you! We again agree. The suggested paper has been added to the references.
Reviewer 2 Report
This is an interesting study providing useful information regarding the role of hybrid immunity and/or vaccination in the COVID-19 vaccine breakthrough. This study reveals that hybrid immunity was more successful than the BNT162b2 mRNA vaccination alone, but the advantage was smaller for risk groups. Both hybrid immunity and the vaccine alone provided less efficient protection against the omicron form, resulting in more outbreaks of infection.
The study was well organized, performed, and statistically analyzed. Despite some limitations (for example, it is an observational study; bias due to different testing frequencies, etc.), the authors' findings are important for public healthcare and fully justified.
Author Response
Reviewer 2
This is an interesting study providing useful information regarding the role of hybrid immunity and/or vaccination in the COVID-19 vaccine breakthrough. This study reveals that hybrid immunity was more successful than the BNT162b2 mRNA vaccination alone, but the advantage was smaller for risk groups. Both hybrid immunity and the vaccine alone provided less efficient protection against the omicron form, resulting in more outbreaks of infection.
Authors’ response: Thank you! We agree and appreciate this comment.
The study was well organized, performed, and statistically analyzed. Despite some limitations (for example, it is an observational study; bias due to different testing frequencies, etc.), the authors' findings are important for public healthcare and fully justified.
Authors’ response: Thank you! We much appreciate this comment.
Round 2
Reviewer 1 Report
The authors have answered all the queries.